# Convergent Canonical Pathways in Autism Spectrum Disorder from Proteomic, Transcriptomic and DNA Methylation Data

**DOI:** 10.3390/ijms221910757

**Published:** 2021-10-05

**Authors:** Caitlyn Mahony, Colleen O’Ryan

**Affiliations:** Molecular & Cell Biology, University of Cape Town, Private Bag, Rondebosch, Cape Town 7700, South Africa; MHNCAI001@myuct.ac.za

**Keywords:** proteomics, transcriptomics, DNA methylation, mitochondria, metabolism, OXPHOS, ASD, neurogenesis, gliosis, neurodevelopment

## Abstract

Autism Spectrum Disorder (ASD) is a complex neurodevelopmental disorder with extensive genetic and aetiological heterogeneity. While the underlying molecular mechanisms involved remain unclear, significant progress has been facilitated by recent advances in high-throughput transcriptomic, epigenomic and proteomic technologies. Here, we review recently published ASD proteomic data and compare proteomic functional enrichment signatures with those of transcriptomic and epigenomic data. We identify canonical pathways that are consistently implicated in ASD molecular data and find an enrichment of pathways involved in mitochondrial metabolism and neurogenesis. We identify a subset of differentially expressed proteins that are supported by ASD transcriptomic and DNA methylation data. Furthermore, these differentially expressed proteins are enriched for disease phenotype pathways associated with ASD aetiology. These proteins converge on protein–protein interaction networks that regulate cell proliferation and differentiation, metabolism, and inflammation, which demonstrates a link between canonical pathways, biological processes and the ASD phenotype. This review highlights how proteomics can uncover potential molecular mechanisms to explain a link between mitochondrial dysfunction and neurodevelopmental pathology.

## 1. Introduction

Autism Spectrum Disorder (ASD) is a lifelong neurodevelopmental condition that is characterized by heterogeneous genetic origins and a poorly understood aetiology [1]. Although the heritability of ASD is undisputed, the genetic contribution of ASD is only 64.6% [2] to 80% [3]. The genetic architecture of ASD comprises highly penetrant, rare variants, as well as more common variants which have smaller effect sizes [4]. However, no single genetic variant accounts for more than 1% of ASD cases [5], with more than 90% of ASD cases being idiopathic [6]. Recent technological advances in genomics, coupled with systems biology approaches that use functional genomic datasets, have facilitated remarkable progress in understanding the functional convergence of the genetic loci associated with ASD [7]. However, a challenge of ASD molecular research is being able to “translate gene discovery into an actionable understanding of ASD pathology” [8]. Transcriptomic and epigenomic approaches are widely used to study gene regulation and expression in ASD, and these have provided some insight into the functional changes at the molecular level. However, there have been fewer proteomic studies in ASD, due in part to the numerous challenges of this approach. Proteomic methodological approaches vary widely with respect to sample preparation, instrumentation, and protein quantification methods, all of which culminates in studies with limited reproducibility [9]. Additionally, the lack of unified data repositories and standardised protein identifiers make it difficult to integrate results between different studies [10]. However, as the “undertakers of biological activities” [11], proteins directly reflect the physiological processes underpinning disease aetiology. Therefore, proteomics is an essential tool to characterise the cellular mechanisms involved in ASD aetiology and to validate the changes observed in the ASD transcriptome and epigenome.

We propose that the integration of ASD proteomic data with large scale ASD transcriptomic and DNA methylation (DNAm) data may yield insight into the molecular mechanisms of ASD. Despite the aetiological and genetic complexity of ASD, this underlying heterogeneity is thought to converge on a limited number of biochemical pathways [12]. Integrative analyses of disparate molecular datasets are well-suited for the identification of common functional networks implicated in complex disorders and these networks can identify potential relationships between disease mechanisms and phenotypes [13]. In this review, we collate recently published data from independent transcriptomic, DNAm, and proteomics studies to identify canonical pathways that are consistently implicated in ASD. Importantly, the purpose of this review is not to combine these disparate datasets in a bioinformatic analysis. Instead, these datasets will be functionally annotated in a uniform manner for comparison. First, we will examine this data to identify common shared canonical pathways across different molecular datasets. Subsequently, we will explore ASD proteomic datasets to identify differentially expressed proteins that are also implicated in the transcriptomic and DNAm data. Finally, these proteins will be characterised with respect to enriched protein–protein interaction networks to investigate the link between differentially expressed proteins, enriched canonical pathways and biological processes involved in ASD aetiology.

## 2. Methods

This review integrated data from large-scale quantitative meta-analyses from both transcriptomic and DNAm data, as well as from ASD proteomic datasets published over the past five years. Overall, data were collated from 19 studies and meta-analyses from ASD cohorts published between 2017 and 2021 (Table 1). The publications included six proteomic datasets [11,14,15,16,17,18], eight transcriptomic studies [19,20,21,22,23,24,25,26], and five DNAm screens [26,27,28,29,30]. The data collated from each study were the published list of genes or proteins found to be significantly associated with ASD after data quality control, normalisation, processing, statistical analyses and peer review (Appendix A). The datasets used in this review were treated as inherently and intentionally heterogeneous given that the data were: (i) derived from different cohorts spanning a wide range of ages and tissue types from both blood and brain samples, (ii) generated using different molecular approaches, (iii) subject to different data processing and quality control workflows, and (iv) filtered using different methods to determine statistical significance. Rather than combine the raw datasets in a bioinformatic analysis, the final published lists of ASD-associated genes from each dataset were functionally annotated and analysed to identify common canonical pathways. The integration of such heterogeneous datasets aimed to highlight canonical pathways that are dysregulated across a range of molecular datasets from different cohorts, tissue types and ages, which could indicate common underlying mechanisms in ASD.

The gene list from each study was converted into NCBI Entrez IDs using a combination of gene ID conversion tools, namely (i) Database for Annotation, Visualization and Integrated Discovery (DAVID) v6.8 Gene Accession Conversion Tool (https://david.ncifcrf.gov/conversion.jsp. Accessed on 8 March 2021) [31,32], (ii) the g:Profiler tool g:convert for Gene ID conversion (https://biit.cs.ut.ee/gprofiler/convert. Accessed on 8 March 2021) [33], (iii) the GIDcon batchwise gene ID conversion tool (http://resource.ibab.ac.in/GIDCON/geneid/home.html. Accessed on 8 March 2021) and (iv) the SYNGO geneset analysis tool (https://syngoportal.org/convert.html. Accessed on 8 March 2021). For the datasets written as Ensemble Gene IDs, the Ensemble ID History Converter (https://www.ensembl.org/Homo_sapiens/Tools/IDMapper. Accessed on 8 March 2021) was used to map all identifiers to matched identifiers in the most recent release of the database. Subsequently, all Ensemble Identifiers, Gene Symbols or GenBank accession numbers not identified through any of these ID conversion tools were manually submitted to NCBI (https://www.ncbi.nlm.nih.gov/gene. Accessed on 8 March 2021) to find the corresponding Gene ID. Deprecated Ensemble IDs that were not matched to an ID in the most recent release were excluded.

Subsequently, each gene list was subjected to gene set enrichment analysis [GSEA] against the Molecular Signatures Database (MSigDB) [34], using the open-source GSEA tool developed by UC San Diego and the Broad Institute (http://www.gsea-msigdb.org/gsea/msigdb/annotate.jsp. Accessed on 12 April 2021) [35,36]. Each dataset was annotated with respect to the top 10 significantly enriched Hallmark Canonical pathways where each pathway represents a well-defined biological process in one curated gene set [37]. These enrichment signatures were compared between independent datasets to identify canonical pathways that were consistently dysregulated in ASD at all three molecular levels: proteomic, transcriptomic and epigenomic. 

The proteomic datasets were examined to identify a subset of differentially expressed proteins that were also implicated in transcriptomic and DNAm data in ASD. After conversion to NCBI Entrez gene IDs, this subset of proteins was annotated with respect to significantly enriched Hallmark canonical pathways, ClinVar disease pathways and transcription factor protein–protein interactions (TF-PPIs) using the Enrichr gene list analysis tool suite (https://maayanlab.cloud/Enrichr/. Accessed on 12 April 2021) [38,39]. TF-PPIs are defined by a library of datasets corresponding to a list of transcription factors and the proteins that interact with them. The top 10 significantly enriched TF-PPIs were used to generate a signalling network based on the SIGnaling Network Open Resource (SIGNOR) v2.0 database [40], using the Network Analyst web interface (https://www.networkanalyst.ca/NetworkAnalyst/home.xhtml. Accessed on 12 April 2021) [41]. This network was used to explore the relationship between the canonical pathways implicated in ASD, biological processes and phenotypic aspects of ASD aetiology.

## 3. Results

After collating the data from 19 different studies (Table 1), we determined the top 10 significantly enriched Hallmark canonical pathways for each of the datasets (Appendix A). These enrichment signatures were compared across all 19 datasets to identify canonical pathways that were consistently supported by all three molecular approaches (Table 2). Eight Hallmark canonical pathways were enriched in seven or more different datasets (Figure 1). Of these, six canonical pathways were supported using all three molecular approaches, with five of these pathways shown to be dysregulated in brain tissue using all three molecular approaches (Appendix A). These were the pathways for oxidative phosphorylation, mTORC1 signalling, coagulation, xenobiotic metabolism and adipogenesis. Notably, three canonical pathways—oxidative phosphorylation, mTORC1 signalling and xenobiotic metabolism- were each implicated in at least three proteomic, three transcriptomic and two DNA methylation studies. The oxidative phosphorylation and mTORC1 signalling pathways were the two pathways that were most frequently associated with ASD. These two pathways were each implicated in 10 different datasets and these datasets included proteomic, transcriptomic and DNAm datasets derived from post-mortem brain tissue from individuals with ASD. While post-mortem brain tissue does not always reflect disease mechanisms that are relevant during neurodevelopment and early childhood, it is important to note that both the mTORC1 and oxidative phosphorylation pathways were also implicated in enrichment signatures from cohorts of younger children. Specifically, both pathways were implicated in blood proteomic profiles in children ranging from 18 months to 8 years [14], frontal cortex transcriptomic signatures from cohorts where the mean age was between 20.3–28 years [23,26] as well as DNAm data from children between 6–12 years [30].

The four additional canonical pathways that were consistently implicated across molecular datasets were the xenobiotic metabolism, p53, adipogenesis and coagulation pathways. Both the xenobiotic metabolism and adipogenesis pathways were implicated in enrichment signatures from brain tissue using all three molecular approaches and were also implicated in studies of younger children using proteomic [14,15], transcriptomic [20,26] and DNAm approaches [26,27,30]. While the p53 pathway was not implicated in proteomic data from brain tissue, five transcriptomic datasets from brain tissue were enriched for p53 signalling [22,23,25,26] as well as three DNAm screens from blood, brain, and buccal tissue [26,28,30].

It is important to note that four of six proteomic datasets included in this review are blood-based datasets, which means that they provide limited insight into which proteins might be differentially expressed in the brain. Therefore, this review compared the canonical pathways implicated in blood-based proteomic profiles with those enriched in proteomic, transcriptomic and DNAm data from brain tissue in order to highlight canonical pathways that could have functional implications in ASD. The coagulation pathway was the canonical pathway that was most consistently enriched in the ASD proteomic data. This pathway was implicated in five out of six proteomic datasets, including three datasets derived from blood samples in children between 18 months and 15 years [14,15,16] as well as post-mortem brain tissue from BA19 and the cerebellum [18]. The coagulation pathway was also enriched in both transcriptomic and DNAm datasets derived from umbilical cord blood [20,27], as well as transcriptomic data from brain tissue in individuals between 24 and 36 years old [17]. The complement pathway, which is a related component of the innate and adaptive immune response pathways, was implicated in three proteomic profiles in blood, and was also dysregulated in three transcriptomic studies in ASD brain tissue [23,24,25]. In addition, the allograft rejection pathway, which can be induced by complement activation, was enriched in blood based proteomic profiles [11] and transcriptomic data from blood [19,22] and brain tissue [22,24].

We also used a second approach to explore how the proteomic data articulated with the transcriptomic and DNAm data associated with ASD. We identified a subset of 121 proteins that were implicated in ASD using all three molecular approaches (Appendix A). Of these proteins, 53 were shown to be altered in ASD brain tissue using all three molecular methods. Of the proteins that were differentially expressed in blood, 10 were also implicated in transcriptomic and DNAm data in brain tissue; 12 were dysregulated in brain transcriptomic profiles and 25 were supported by DNAm studies in brain tissue. The remaining 21 proteins were shown to be altered in blood using proteomic, transcriptomic and DNAm screens. We characterised the dataset of all 121 differentially expressed proteins that were supported by transcriptomic and DNAm data with respect to enriched Hallmark canonical pathways, disease phenotype pathways, and downstream biological processes in order to explore the functional implications associated with these proteins. This subset of proteins was significantly enriched for Hallmark canonical pathways involved in mTOR signalling (this includes mTORC1 signalling and PI3K-AKT-mTOR signalling), metabolism (this includes oxidative phosphorylation, glycolysis, fatty acid metabolism and adipogenesis), and immune responses (this includes complement, allograft rejection and IL6-JAK-STAT3 signalling) (Appendix A). The 10 most significantly enriched Hallmark canonical pathways in this subset of proteins included four canonical pathways that were also consistently enriched across all 19 datasets: mTORC1 signalling, oxidative phosphorylation, adipogenesis and the complement response (Figure 2A). Together, the four canonical pathways highlighted in both analyses converge on signalling networks that regulate neural stem cell proliferation, differentiation, metabolism, redox homeostasis, and reactive gliosis during neurodevelopment (Figure 2B).

The subset of 121 proteins that were implicated using all three molecular approaches was also enriched for five ClinVar disease pathways associated with neurological, immunological, and metabolic diseases (Table 3). Notably, the most significantly enriched disease pathway was for Leigh syndrome, which is a paediatric mitochondrial disease that manifests with severe neuropathology [42]. Therefore, the above-mentioned proteins converge on three central components of ASD pathophysiology which may yield insight into the link between canonical pathways and the dysregulation of biological processes in ASD aetiology. Consequently, we examined the protein–protein interaction networks and downstream signalling pathways associated with this dataset by testing for significant enrichment of TF-PPIs. The top 10 significantly enriched TF-PPIs are key transcriptional regulators of mitochondrial metabolism (CREBP1A, PPARGC1A and HNF1A), lipid metabolism (PPARCG1A and STAT1), adipogenesis (NR3C1), the p53 pathway (TP53; TP63) and inflammation (NR3C1, STAT1) (Appendix A). Notably, several of these transcription factors mediate signalling via the PI3K–AKT–mTOR (HTT, TP53, CREBP1A, PPARGC1A) or ERK–mTOR (ESR1, STAT1, HNF1A) pathways. This highlights some of the transcription factors that regulate the canonical pathways implicated in ASD molecular data. The signalling network between the top 10 significantly enriched transcription factors converged on two SIGNOR stimuli (ROS and DNA damage), as well as six SIGNOR signalling pathways, namely; polarization, proliferation, apoptosis, cell death, mitochondrial biogenesis, and inflammation (Figure 3). Therefore, this subset of differentially expressed proteins highlights a link between signalling pathways, biological processes and ClinVar disease phenotypes associated with ASD aetiology. Altogether, recent proteomic profiling studies, in conjunction with previously published transcriptomic and epigenomic meta-analyses, consistently implicate canonical pathways involved in neuronal metabolism, differentiation and inflammation.

## 4. Discussion

Molecular research of ASD requires a “multi-omics” approach to comprehensively characterise the convergent biological processes associated with its complex genetic architecture. Proteomic, transcriptomic, and epigenomic approaches are well-established as ways to investigate changes in gene regulation, expression, and function in ASD. DNA methylation is widely recognized as an epigenetic regulator of gene expression that contributes to ASD aetiology [43]. Transcriptomic studies facilitate an understanding of the link between the genetic mutations associated with ASD and changes in gene expression and function [11]. Recent advances in high-throughput proteomic technologies have contributed towards the progress of ASD proteomic profiling studies using both peripheral and brain tissue to provide insight into the cellular mechanisms involved in ASD aetiology. Proteomic approaches come with a unique set of advantages and limitations in the context of ASD research. In contrast to other molecular methods, proteomic studies are able to detect different protein isoforms or post-translational modifications and have the potential to identify disease biomarkers and drug targets for the diagnosis and treatment of ASD [9,11]. One major challenge associated with the identification of proteomic biomarkers in ASD is the scarcity of brain tissue available for research, coupled with the fact that brain tissue is generally inaccessible as the tissue of choice for diagnosis. Recent reviews of ASD proteomic data highlight the utility of blood-based proteomic biomarkers in ASD; the proteomic expression profiles of peripheral tissues have been shown to correlate moderately with those of brain tissue, and blood samples are more easily accessed and less invasive to obtain for diagnostic purposes [11]. Additionally, novel bioinformatic approaches have been used to predict ASD-associated blood secretory proteins; and this approach has recently been experimentally validated in an ASD cohort [17]. Nevertheless, the variation in protein expression between different tissue types remains an important consideration for the analysis of proteomic datasets. This is compounded by the fact that protein expression can also vary widely between different brain regions, which means that even brain-based proteomic profiles only provide limited insight into disease mechanisms. This highlights how the aetiological complexity of ASD is compounded by the heterogeneity between molecular datasets derived from different tissue types, brain regions and molecular methods.

Despite these limitations, proteomics can be an essential component of a multi-omic molecular approach to ASD aetiology. Protein expression studies can be used to investigate biologically relevant protein interaction networks and signalling pathways [18] and can yield insight into the link between molecular and phenotypic aspects of disease aetiology. The approach taken in this review considers both the strengths and limitations of ASD proteomics research. Our review includes proteomics data from both the cortex (BA19) and cerebellum, and we cross-reference these datasets against transcriptomic and DNAm data from predominantly the prefrontal cortex and temporal cortex. These brain regions are relevant in the context of ASD because previous studies have found that ASD-associated changes in mRNA expression, miRNA expression, DNA methylation, and histone acetylation were predominantly localized to the cerebral cortex [26]. This review integrated the functional annotations of ASD proteomic, transcriptomic and DNAm datasets from both blood and brain tissue to identify common canonical pathways associated with ASD. 

We found that the canonical pathways involved in metabolism, redox homeostasis, inflammation and proliferation are consistently supported by proteomic, transcriptomic and DNAm data in ASD (Figure 1 and Figure 2). This review explored the link between canonical pathways, biological processes and disease phenotype pathways in ASD by functionally annotating a subset of differentially expressed proteins that were also supported by transcriptomic and DNAm data. This subset of proteins was enriched for four canonical pathways that were also implicated in seven or more independent datasets. These were the pathways for oxidative phosphorylation, mTORC1 signalling, adipogenesis and the complement response (Figure 2A). Additionally, the other six most significantly enriched Hallmark pathways in this subset of proteins also converge on biological processes implicated across all 19 independent datasets. Both the oestrogen response and PI3K-AKT-MTOR pathways converge on mTORC1 signalling, while the allograft rejection response has been highlighted as a component of the complement cascade, and fatty acid metabolism and glycolysis share common upstream regulators with adipogenesis and oxidative phosphorylation. The ten most significantly enriched TF-PPIs in this subset of proteins highlighted key transcriptional regulators of these canonical pathways. These transcriptional regulators form a signalling network that highlights how these canonical pathways regulate biological processes involved in neuropathology (Figure 3). This network converges on the SIGNOR pathways for mitochondrial biogenesis and inflammation, and this is consistent with prior evidence linking mitochondrial dysfunction [44,45,46,47,48,49] and neuroinflammation [50,51,52] to ASD. The polarization and proliferation pathways are essential regulators of neural stem cell self-renewal and differentiation [53,54,55], while the apoptosis and cell death signalling pathways are implicated as mechanisms whereby microglia drive synaptic pruning, and the maturation and migration of neuronal progenitors [56]. In addition, this subset of proteins was significantly enriched for five ClinVar disease pathways associated with neuropathology, mitochondrial dysfunction or auto-immune disease (Table 3). Therefore, the differentially expressed proteins highlighted in this review are linked to both molecular and phenotypic facets of ASD aetiology. 

The common canonical pathways highlighted above are each established as molecular mechanisms contributing to ASD aetiology. In addition, in vitro and in vivo model systems have demonstrated that these pathways play an essential role in the regulation of neurodevelopment, and converge on a link between mitochondrial dysfunction, neurogenesis, and inflammation. Notably, the functional enrichment signatures of most of the ASD omics datasets included in this review are not representative of early embryogenesis, due to the challenges associated with accessing relevant tissue samples or databanks. Therefore, the datasets included in this review provide limited insight into the mechanisms that might be dysregulated during neurodevelopment. However, this review highlights canonical pathways that are dysregulated in functional enrichment signatures obtained using three different molecular approaches from a wide range of tissue types and individuals of different ages. Therefore, these common pathways may point to common underlying mechanisms in ASD, which has potential relevance in the context of neurodevelopment. While this discussion does not aim to demonstrate a direct link between these molecular mechanisms and disease aetiology, it is useful to consider how the canonical pathways implicated in this review are established as aspects of ASD aetiology, and to review the implications of the dysregulation of these pathways during neurodevelopment.

Firstly, a role for mTORC1 signalling is well described in ASD [57,58]. Mutations in tuberous sclerosis complex (TSC) and phosphatase and tensin homolog (PTEN), which are both central regulators of mTORC1 signalling, are associated with autistic behaviours [59,60,61]. Aberrant mTOR signalling is associated with altered synaptogenesis and ASD-like neurophysiology in animal and organoid model systems [62,63,64,65,66,67]. Moreover, the mTOR signalling pathway is a promising therapeutic target in ASD [68]. The canonical pathways for mTORC1 signalling and oxidative phosphorylation were the two most commonly implicated pathways in our analyses which is consistent with evidence showing that mitochondrial dysfunction is involved in ASD aetiology [44,45,46,47,48,49]. Mitochondrial function has recently been proposed as a central driver of neuronal differentiation [69,70,71,72] and the tight coregulation of mTORC1 signalling and mitochondrial metabolism is essential to control neurogenesis. The transition from undifferentiated neuronal stem cells (NSCs) to mature neurons relies on a metabolic shift from aerobic glycolysis to mitochondrial respiration. The latter fuels neuronal migration, dendrite formation and synaptogenesis [73]. The mTOR signalling pathway plays a central role in driving this metabolic switch, which is essential for neuronal survival and differentiation [71]. Neurogenic factors induce AKT-MTORC1 signalling, which upregulates peroxisome proliferator-activated receptor-gamma coactivator (PPARGC1A) signalling to induce mitochondrial respiration and biogenesis [70,74]. Conversely, cellular metabolism regulates mTOR signalling via the AMP-dependent protein kinase (AMPK), which is inhibited by a decrease in the AMP:ATP ratio [75]. The dysregulation of signalling between mTORC1 and mitochondrial metabolism during development has profound consequences for NSC commitment and differentiation, and the development and maintenance of mature neuronal networks. 

We also observed a consistent enrichment of the adipogenesis pathway in ASD molecular data, which implicates the same networks that regulate stem cell metabolism and differentiation. Adipogenesis is regulated by the “opposite interplay” between WNT/B-catenin signalling and peroxisome proliferator-activated receptor gamma (PPARGy), each of which negatively regulates the other [76]. WNT signalling is directly involved in regulating NSC maintenance, proliferation, and differentiation [77,78] and WNT responsive genes maintain aerobic glycolysis in NSCs [71]. This highlights that the WNT pathway is at the intersection between stem cell metabolism and development. Crosstalk between WNT signalling and mTORC1 signalling is linked to the “Warburg effect” that induces aerobic glycolysis and over-proliferation in cancer cells [76]. A dysregulation of these same pathways during neurodevelopment could disrupt NSC commitment and differentiation. In fact, chronic activation of WNT signalling altered mTOR signalling in human organoids, leading to increased NSC proliferation, impaired neuronal differentiation and disrupted radial glial organization [79]. Moreover, in vivo downregulation of WNT signalling leads to premature neurogenesis and atypical behaviours [80]. The WNT/B catenin pathway has also been implicated in ASD genetic and transcriptomic data [81,82,83,84,85].

On the other hand, PPARGy acts as a master regulator of lipid metabolism and regulates target genes necessary for differentiation, fatty acid transport, carbohydrate metabolism, and energy homeostasis [86]. Mitochondrial fatty acid oxidation (FAO) is essential for the self-renewal of NSCs, but fatty acid metabolism shifts towards lipogenesis during neurogenesis [87,88,89,90]. In vitro studies show that this shift in fatty acid metabolism is regulated by the AMPK-PPARGC1A axis [90]. The coregulation between PPARGy and PPARGC1A means that glucose and fatty acid metabolism are intrinsically linked during neurodevelopment. Disrupting FAO inhibits stem cell self-renewal [88,89] and impairs NSC differentiation in mouse models [91]. Recent reviews of ASD proteomic data also find that proteins involved in lipid metabolism are differentially expressed in ASD [92,93]. A role for PPARGy is supported by increasing evidence of mitochondrial FAO deficiencies in ASD, and PPARGy agonists have been proposed as therapeutic agents in ASD [94,95]. This highlights how PPARGy is implicated in neurodevelopment and ASD, and how the canonical pathways for mTORC1 signalling, oxidative phosphorylation and adipogenesis converge on the signalling between neuronal metabolism and differentiation.

A dysregulation of the signalling between mitochondrial metabolism and neurogenesis has significant implications for neuronal development and function. Firstly, mitochondrial oxidative phosphorylation is one of the primary producers of intracellular reactive oxygen species (ROS) and disruptions to mitochondrial metabolism can lead to oxidative stress [96]. Increased oxidative stress is a well-documented aspect of ASD pathophysiology; the evidence for markers of oxidative stress associated with glutathione metabolism, lipid peroxidation, protein oxidation, DNA oxidation and antioxidant enzyme activity has been comprehensively reviewed in recent years [45,97,98,99,100,101,102]. In the context of neurodevelopment, this has significant implications for the redox regulation of neurogenesis. Undifferentiated NSCs are characterized by high levels of endogenous ROS, which plays a functional role in stem cell maintenance [103,104] and oxidative stress is known to promote NSC self-renewal and inhibit downstream neurogenesis [96,104]. Studies on in vitro and in vivo model systems consistently demonstrate that increasing oxidative stress [105,106,107] or disrupting mitochondrial homeostasis and function [108,109,110,111] leads to an inhibition of neurogenesis and a shift towards gliosis; the latter is associated with neuroinflammation and is established as a hallmark of ASD aetiology. Clinical studies also report that gliosis is one of the neuropathological manifestations of mitochondrial disease [43,112,113,114,115].

Many of the canonical pathways highlighted in this review are implicated as mechanisms involved in the response to oxidative stress and gliosis during neurogenesis. Clinical data has reported mTOR signalling as a mechanism that connects mitochondrial disease to gliosis [116], with both in vitro [117], and in vivo, studies [118] showing that gliosis can be induced by targeted disruptions to mTOR signalling. Crosstalk between the PPARGy and WNT canonical pathways is also associated with increased oxidative stress and chronic inflammation in cancer [119], and WNT/B-catenin-PPARGy signalling can be targeted to reduce reactive gliosis in models of neurodegenerative disease [120,121,122]. In addition, the p53 pathway and xenobiotic metabolism were each implicated in eight or more independent datasets in our analysis and these pathways were supported by all three types of molecular data (Figure 1). The p53 signalling pathway is a central regulator of inflammation, oxidative stress, and apoptosis [123], while the canonical pathway for xenobiotic metabolism is composed of genes that respond to inflammation, metabolic stress, and ROS [124]. Notably, both mTOR and WNT signalling regulate p53 degradation [73,125,126]. The p53 signalling pathway also plays an important role in neural precursor cell self-renewal, neuronal commitment, and reactive gliosis in response to mitochondrial dysfunction [110,111,127]. 

Gliosis is characterized by highly reactive microglia and astrocytes, leading to an overproduction of glial-specific fibrillary acidic protein (GFAP) and inflammatory cytokines [128]. This review highlights the consistent enrichment of inflammatory canonical pathways in ASD transcriptomic and proteomic data. The canonical pathways involved in inflammatory immune response pathways were those most frequently associated with the transcriptomic enrichment signatures: six of ten hallmark canonical pathways that were implicated in at least five independent transcriptomic datasets were related to immune responses. This is consistent with a substantial body of work implicating neuroinflammation and gliosis as mechanisms of pathology of ASD, which has been thoroughly reviewed elsewhere [50,51,52,129,130,131,132]. A role for inflammation, astrocyte function and microglial activation is well-supported by transcriptomic and proteomic studies in ASD brain tissue [133,134,135]. This is also consistent with immunohistochemistry, positron emission tomography and morphological data in ASD [129,136,137]. Notably, microglial phagocytosis is responsible for the degradation of proliferating NSCs during neurodevelopment [57]. This is essential for synaptogenesis and post-natal synaptic pruning, both of which are thought to be dysregulated in ASD [129,138,139]. Importantly, microglial metabolism is closely linked to neuroinflammatory states [140,141], and both glucose and lipid metabolism regulate microglial activation [142,143,144,145]. The dysregulation of glial and microglial metabolism, proliferation and function can alter neuronal differentiation, impair synaptogenesis and pruning, and change neuroinflammatory states, with profound implications for neural architecture and connectivity [146]. 

Our analysis of recent ASD proteomic profiles further supports a role for neuroinflammation in ASD. The complement and coagulation cascades were implicated in three and five out of six ASD proteomic studies respectively, which is supported in independent reviews of ASD proteomic data [13,92,93]. Both cascades form part of the innate immune system and interactions between them via mannose-binding lectin-associated serine proteases are well documented [147,148,149,150,151,152]. Both cascades are involved in the inflammatory response by reactive microglia [153,154]. Coagulation proteins affect the morphology, proliferation, and function of astrocytes [153], while the complement system is implicated as a mechanism by which microglia regulate neurogenesis, neuronal migration and synaptic pruning [154,155,156,157]. Complement proteins play a role in microglial activation, which can influence neuroinflammatory signalling and neurodevelopment [158]. Animal models show that genetic knockdown of complement proteins impairs neuronal migration, while activation of the complement system rescues this deficit [155]. There is mounting evidence for complement system dysfunction in neurodevelopmental disorders [156], which has been proposed as one of the mechanisms behind the synaptic pruning deficits, increased dendritic spine density, cortical hyperconnectivity and resultant behavioural phenotypes in ASD [157]. Therefore, the immune response cascades enriched in ASD proteomic profiles function as a link between gliosis, neurogenesis, and synaptogenesis, highlighting a point of convergence between these different mechanisms in ASD pathology.

Collectively, the canonical pathways highlighted in this review are consistent with known pathways that to contribute to ASD aetiology, as well as pathways that have been implicated in independent reviews of ASD molecular data. Importantly, our review also considers the interactions between these canonical pathways, particularly in the context of neurodevelopment. The latter is often overlooked when the relevance of each canonical pathway is evaluated in isolation. Previous reviews have comprehensively described the role of mTORC1 signalling in protein synthesis, neuronal differentiation, migration, and patterning [57], but these reviews have focused less on the role of mTORC1 as a regulator of mitochondrial metabolism. The same is true for reviews on the role played by WNT signalling in neurogenesis and ASD [159,160], which describe the central role played by WNT signalling in NSC proliferation and differentiation. However, our review emphasises how WNT-PPARGy signalling intersects with PPARGC1A, anti-inflammatory and antioxidant responses, and the regulation of mitochondrial biogenesis and metabolism. Given that mitochondrial dysfunction is an established component of ASD aetiology, and that metabolic state plays a central role in neuronal differentiation, our review highlights a potentially important connection between mTORC1, oxidative phosphorylation and neuropathology in ASD. Finally, our review integrates all of the above with the thoroughly reviewed evidence for neuroinflammation and oxidative stress in ASD [49,50,51,52,97,98,99,100,101,102] by considering the role of mitochondrial metabolism in gliosis and microglial activation, and the downstream consequences for synaptogenesis, neuronal connectivity and behavioural phenotypes. Therefore, our review is not only consistent with current molecular mechanisms implicated in ASD, but also highlights an under-explored link between mitochondrial dysfunction, neuroinflammation and neurodevelopmental pathology. 

## 5. Conclusions

The genetic and phenotypic complexity of ASD makes it challenging to identify the molecular mechanisms that contribute to its aetiology. Proteomic approaches can yield novel insight into the dysregulation of biological processes and signalling networks in ASD. Despite the power of proteomics, there are currently limitations to this approach, both in terms of sample acquisition, processing, and analysis, and in accurately capturing variation between different tissue types and developmental stages. In light of these caveats, we integrated recently published ASD proteomic profiles with large-scale meta-analyses of whole-genome transcriptomic and DNAm data to compare the functional enrichment signatures of disparate molecular datasets. Our analysis demonstrates that ASD proteomic, transcriptomic and DNAm data consistently support the dysregulation of mTORC1 signalling, oxidative phosphorylation, adipogenesis and the response to inflammation and oxidative stress. The mTORC1-, WNT- and PPARGy-signalling pathways are each well-established regulators of neurodevelopment that are also implicated in ASD and neurodevelopmental pathology. The mTORC1 signalling pathway regulates oxidative phosphorylation via PPARGC1A, which operates in concert with PPARGy to regulate mitochondrial respiration and fatty acid oxidation. These pathways play an essential role in the tight coupling between stem cell proliferation, differentiation, and mitochondrial metabolism during neurogenesis. If this coupling is disrupted, this can induce metabolic and oxidative stress in NSCs. Both metabolic and oxidative stress are known to disrupt neurogenesis, promote gliosis and induce microglial activation. This proposes a link between mitochondrial dysfunction, oxidative stress, and gliosis, each of which are well-established features of ASD aetiology. Given the central role played by glial networks in neuronal migration and synaptogenesis, gliosis is implicated as a mechanism involved in neuroinflammation and the neuropathophysiology associated with ASD. In summary, this review demonstrates that ASD molecular data converges on canonical pathways involved in mitochondrial function, neurogenesis and neuroinflammation, highlighting how these three key aspects of ASD aetiology interact, and the relevance of these interactions in the context of neurodevelopment.

## Figures and Tables

**Figure 1 ijms-22-10757-f001:**
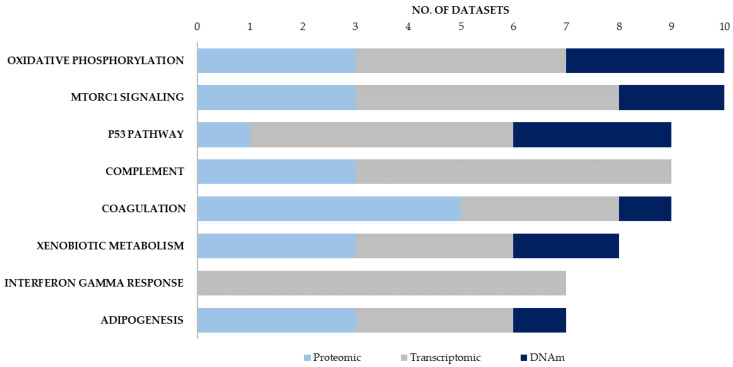
Hallmark canonical pathways most consistently implicated in ASD molecular data. Datasets were collated from 19 studies and meta-analyses on ASD papers published between 2017 and 2021: six proteomic, eight transcriptomic and six DNA methylation (DNAm) studies. All gene lists were converted into NCBI Entrez gene IDs. Each dataset was annotated with respect to the top 10 significantly enriched Hallmark canonical pathways. The Hallmark canonical pathways that were significantly enriched in seven or more independent ASD proteomic, transcriptomic and DNAm datasets are shown; those shown in bold are those consistently implicated using all three molecular approaches.

**Figure 2 ijms-22-10757-f002:**
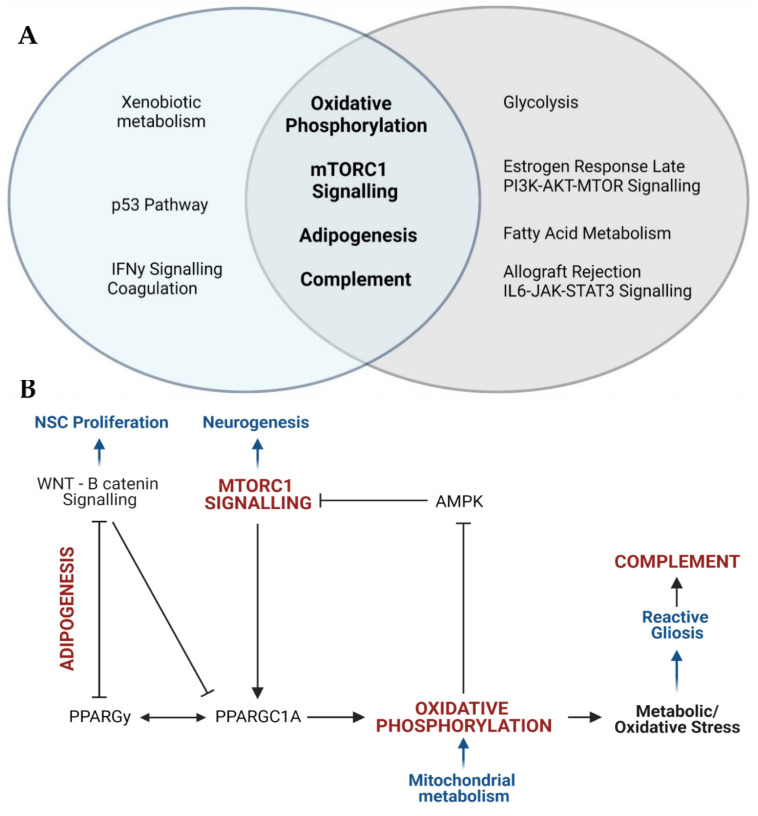
ASD proteomic profiles converge on four canonical pathways involved in mitochondrial metabolism, neurogenesis and neuroinflammation. (**A**) The Venn diagram shows the Hallmark canonical pathways implicated in seven or more independent datasets (in blue), the enriched canonical pathways in the 121 differentially expressed proteins implicated in transcriptomic and DNAm datasets (light grey) and the pathways that overlap (in bold font). (**B**) The four pathways implicated in both analyses (Figure 2A) converge on the regulation of the following biological processes (in blue): neural stem cell proliferation, neurogenesis, mitochondrial metabolism, and inflammation. The mTORC1 signalling pathway induces oxidative phosphorylation via the activation of camp-responsive element binding protein 1 (CREB1) and peroxisome proliferator-activated receptor gamma coactivator 1-alpha (PPARGC1A), which regulates mitochondrial metabolism and the antioxidant response. PPARGC1A is regulated by peroxisome proliferator-activated receptor gamma (PPARGy), which acts as a master regulator of lipid metabolism and is essential to induce adipogenesis. Adipogenesis is negatively regulated by WNT signalling, which inhibits PPARGy, PPARGC1A and mTORC1 signalling to regulate stem cell proliferation and differentiation. Mitochondrial metabolism regulates metabolic and redox homeostasis which governs the inflammatory profile of microglia. Oxidative stress leads to the over proliferation of reactive microglia, which triggers the pro-inflammatory complement response.

**Figure 3 ijms-22-10757-f003:**
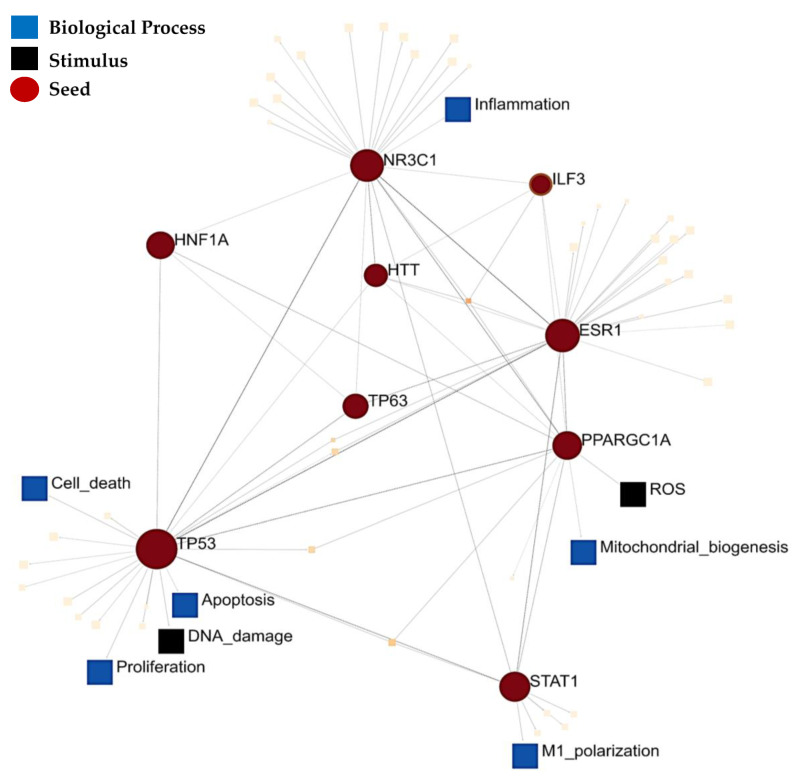
Signalling network between the top 10 transcription factor protein–protein interactions (PPI) enriched in the set of 121 proteins that are supported by ASD proteomic, transcriptomic, and epigenomic datasets. This PPI network was annotated with respect to signalling pathways in the SIGnaling Network Open Resource (SIGNOR) v2.0 database of manually annotated causal relationships between proteins that participate in signal transduction.

**Table 1 ijms-22-10757-t001:** A description of the 19 datasets included in this review, summarizing the type of study, the cohort size, the tissue type, and the definition of the final dataset used from each study. DE = differentially expressed; DM = differentially methylated.

Proteomics Analysis				
Shen et al., 2019 [11]	41 DE proteins in ASD	24 male and 6 female ASD (2–6 yrs) and age/gender-matched controls	Blood	Blood PBMCs
Hewitson et al., 2021 [14]	86 downregulated, 52 upregulated proteins in ASD (FDR < 0.05)	76 ASD (boys) and 78 controls (boys), 18 months to 8 yrs		Blood serum
Shen et al., 2018 [15]	24 DE proteins in ASD	24 male and 6 female autistic patients (2–6 yrs) and age/gender-matched controls		Blood plasma
Yang et al., 2018 [16]	Eight biomarker peaks with higher expression in ASD	Han Chinese children: 68 ASD (average age = 12.4 yrs) and 80 age-matched controls (average age 14.3 yrs)		Blood serum
Yao et al., 2021 [17]	59 genes predicted to encode ASD-related blood-secretory proteins; six proteins were validated using an ELISA	79 brain tissue samples from 19 ASD and 17 controls; ELISA analysis of 20 ASD, and 20 age/gender-matched controls. The average age of the patients and controls were 24 yrs (ranged from 2 to 56) and 34.6 yrs (ranged from 16 to 56) respectively	Brain tissue	Blood samples
Abraham et al., 2019 [18]	146 DE proteins from BA19 between ASD and controls (*p* < 0.05)	9 ASD cases (2–60 yrs) and 9 age- and gender-matched controls (1–60 yrs)		Cerebellum (CB) and Brodmann area 19 (BA19)
**Transcriptomic Meta-Analysis**				
**Reference**	**Dataset Used in Analysis**	**Cohort**	**Tissue**	
Tylee et al., 2017 [19]	90 DE genes in ASD (*p* < 0.05)	626 ASD and 447 controls across seven independent studies; mean age and SD of 5 ± 3.8 yrs	Blood	Ex vivo peripheral blood samples or isolated leukocyte samples derived from peripheral blood
Mordaunt et al., 2019 [20]	172 DE genes in ASD (*p* < 0.01)	59 ASD, 92 non-typically developing, 120 typically developing controls		Umbilical cord blood samples from both the Markers of Autism Risk in Babies-Learning Early Signs (MARBLES) and the Early Autism Risk Longitudinal Investigation (EARLI) high-risk pregnancy cohorts
Gao et al., 2020 [21]	3624 DE genes in ASD	96 ASD and 42 controls age range: 2–18 yrs		Peripheral blood samples (GSE18123 and GSE6575)
He et al., 2019 [22]	DE genes (*p* < 0.05) in ASD	485 ASD and 398 controls		Five data sets from blood lymphoblastoid cell lines (LCLs) (GSE18123, GSE25507, GSE29691, GSE37772, GSE42133)
He et al., 2019 [22]	DE genes (*p* < 0.05) in ASD	109 ASD and 129 controls	Brain tissue	Three data sets from postmortem brain tissue (GSE28475, GSE28521, GSE38322)
Forés-Martos et al., 2019 [23]	1055 DE genes in ASD (FDR *p* < 0.05)	34 ASD cases and 130 controls across three studies. Mean age 20.3 yrs		Frontal cortex tissue
Rahman et al., 2020 [24]	1567 DE genes in ASD	15 ASD and 15 controls across two studies		Post-mortem brain tissue (GSE30573 and GSE64018)
Wright et al., 2017 [25]	1463 DE genes in ASD across all moderately expressed Ensembl genes (13,011) at marginal statistical (*p* < 0.05) significance	13 ASD (3F, 10M), average age 22 yrs (4 to 67) and 39 controls (9F, 30M), average age 22 yrs (2 to 69)		Postmortem brain tissue: dorsolateral prefrontal cortex
Yao et al., 2021 [17]	364 DE genes in ASD	79 brain tissue samples from 19 ASD and 17 controls. Average age of the patients and controls were 24 yrs (2 to 56) and 34.6 yrs (16 to 56) respectively		Brain tissue: cerebellum, frontal cortex, and temporal cortex (GSE28521)
Ramaswami et al., 2020 [26]	5200 DE genes (FDR < 0.05)	82 ASD samples and 74 control samples from 47 ASD and 44 control brains from (Parikshak et al.); mean age and SD were 28 (+/−17) yrs		Frontal and temporal cortex tissue from the Harvard Autism Tissue Program and NIH Neuro Brain Bank
**DNA Methylation Analysis**				
Mordaunt et al., 2020 [27]	537 DM genes in both discovery and replication sets in males	Discovery set = 74 males (35 ASD and 39 controls) and 32 females (15 ASD and 17 controls) in the MARBLES and EARLI studies. Replication set = 38 males (21 ASD and 17 controls) and 8 females (5 ASD and 3 controls)	Blood	Umbilical cord blood samples
Hu et al., 2020 [28]	181 DM genes that overlap between the discovery and validation groups MALES	21 ASD and 21 controls, average age 8.4 yrs		LCLs
Wong et al., 2019 [29]	i)Top ranked iASD-associated DM probes identified in the cross-cortex model incorporating both prefrontal cortex and temporal cortex data	43 ASD and 38 controls, average age at death 29.0 (+/−18.9) and 48.7 (+/−8.8) yrs respectively	Brain tissue	Post-mortem brain tissue from prefrontal cortex, temporal cortex and cerebellum
Ramaswami et al., 2020 [26]	DM genes (promoter or gene body; FDR < 0.05)	56 ASD samples and 41 control samples from 33 ASD and 26 control brains. Mean age and SD = 34 (+/−15) yrs		Frontal and temporal cortex tissue from the Harvard Autism Tissue Program and NIH Neuro Brain Bank
Stathopoulous et al., 2020 [30]	898 DM genes in ASD	48 boys (32 ASD and 16 controls, 6–12 yrs)	Buccal cells	Buccal DNA

**Table 2 ijms-22-10757-t002:** A summary of the Hallmark Enrichment Signatures of 19 ASD molecular datasets, highlighting canonical pathways that were most consistently implicated in ASD proteomic, transcriptomic, and DNAm datasets from blood and brain tissue. Each independent dataset was annotated with respect to the top ten significantly enriched Hallmark canonical pathways (FDR < 0.5); data indicate the number of enrichment signatures implicating each canonical pathway. Pathways in bold = strongly supported using all three molecular approaches; pathways in italicised = supported by two (of three) molecular datasets.

No. of Enrichment Signatures					
Proteomic		Transcriptomic		DNAm	
Hallmark Canonical Pathways	Blood/Brain	Hallmark Canonical Pathways	Blood/Brain	Hallmark Canonical Pathways	Blood/Brain/Buccal
COAGULATION	3/2	INTERFERON GAMMA RESPONSE	3/6	**OXIDATIVE PHOSPHORYLATION**	0/2/1
*COMPLEMENT*	3/0	COMPLEMENT	4/4	*P53 PATHWAY*	1/1/1
**OXIDATIVE PHOSPHORYLATION**	1/2	**MTORC1 SIGNALING**	3/3	MITOTIC SPINDLE	1/1/1
**MTORC1 SIGNALING**	1/2	P53 PATHWAY	2/5	**MTORC1 SIGNALING**	0/1/1
*XENOBIOTIC METABOLISM*	1/2	ALLOGRAFT REJECTION	3/3	*XENOBIOTIC METABOLISM*	1/0/1
ADIPOGENESIS	1/2	INTERFERON ALPHA RESPONSE	4/4	UV RESPONSE UP	1/1/0
*UNFOLDED PROTEIN RESPONSE*	1/1	TNFA SIGNALING VIA NFKB	4/3	*UNFOLDED PROTEIN* *RESPONSE*	0/2/0
MYOGENESIS	1/1	HYPOXIA	2/3	ESTROGEN RESPONSE EARLY	1/1/0
FATTY ACID METABOLISM	1/1	INFLAMMATORY RESPONSE	3/4	E2F TARGETS	0/1/1
MYC TARGETS V1	1/1	APOPTOSIS	3/4	DNA REPAIR	0/1/1
ANGIOGENESIS	2/0	**OXIDATIVE PHOSPHORYLATION**	1/3	PEROXISOME	0/2/0
		EPITHELIAL MESENCHYMAL TRANSITION	1/4		
		KRAS SIGNALING UP	0/4		

**Table 3 ijms-22-10757-t003:** ClinVar disease pathways that are significantly enriched in the 121 differentially expressed proteins that are also implicated in both transcriptomic and DNAm datasets.

ClinVar Disease Pathway	Disease Phenotype	*p* Value
Leigh Syndrome	Neurological disease	0.0027
Familial Partial Lipodystrophy	Metabolic disease	0.0299
Pyruvate Dehydrogenase Complex Deficiency		0.0299
Mitochondrial DNA Deletion Syndrome		0.0474
Autoimmune Lymphoproliferative Syndrome	Autoimmune disease	0.0358

## Data Availability

The data presented in this study are available within the article and Appendix A here.

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
