# Peer review of "Convergent Canonical Pathways in Autism Spectrum Disorder from Proteomic, Transcriptomic and DNA Methylation Data"

_ijms, 2021, doi:10.3390/ijms221910757_

Round 1
Reviewer 1 Report
The manuscript by Mahony and O'Ryan is categorized to a review manuscript, but it is indeed a research manuscript since datasets from multiple studies were compared to report newer conclusions.
I have several comments if this is a research article.
- Since blood proteins are produced from all types of cells in our body, DEPs are not necessarily secreted from regions in brain. In this regard, blood data can be useful for the diagnostic purpose. Thus, it would be great to find a list of potential biomarkers for ASD. Then newer issue will be how the authors can validate the findings.
- To understand a underlying mechanism of ASD, data from tissues can be used. Trying connecting blood components to disease mechanisms could mislead the outcome. Thus, the authors might have to separate tissue/cell data from blood data for this purpose.
- In case where this manuscript is still within the review article, I suggest to delete all new result of re-analysis done by the authors and to summarize points to deliver to the readership. (I guess their message is within 'canonical pathways' in ASD.)
Author Response
We thank the reviewer for the constructive comments. We address each of the three comments below:
Reviewer Comment 1: Since blood proteins are produced from all types of cells in our body, DEPs are not necessarily secreted from regions in brain. In this regard, blood data can be useful for the diagnostic purpose. Thus, it would be great to find a list of potential biomarkers for ASD. Then newer issue will be how the authors can validate the findings. 
The authors agree with the statements raised above by the reviewer, i.e. that DEPs are not necessarily secreted in the brain; that DEP from blood can be useful for diagnostic purposes and that it would be great to identify potential biomarkers for ASD. In our manuscript, we reference several recent reviews of ASD proteomic data, which focus specifically on the identification of biomarkers using proteomic approaches [see lines 44-49; 286-297 and references 9,10,12,14,17], however this was not the focus of our own review. We have amended our manuscript to be more explicit about the limitations associated with blood-based proteomics in the context of neurodevelopmental disorders in our manuscript [see lines 165-170; 294-299; 348-352;524-526].
  Reviewer Comment 2: To understand a underlying mechanism of ASD, data from tissues can be used. Trying connecting blood components to disease mechanisms could mislead the outcome. Thus, the authors might have to separate tissue/cell data from blood data for this purpose.
The authors recognize that connecting data from blood samples to disease mechanisms could be misleading. We have qualified statements in our manuscript where a direct connection may be implied. We have added a paragraph in the results that describes in greater detail which canonical pathways were implicated in brain tissue, which methods were used and the age of the relevant cohorts [see lines 142 – 183]. We have updated Table 2 and Supplementary Figure S2 to include the number of enrichment signatures from each tissue type that supports each canonical pathway. When integrating the datasets from blood and brain tissue we sought to identify which canonical pathways that were implicated in blood samples were indeed also dysregulated in brain tissue. This demonstrates that the canonical pathways that we focus on, are those pathways that are supported in multiple different tissue types, including brain tissue, using multiple different methods across various different cohorts. Additionally, we have amended Supplementary Table S2 to clearly demarcate which proteins were directly implicated in brain tissue, which proteins are supported by transcriptomic and DNAm data from brain tissue, and which proteins were not supported by data from brain tissue [see lines 199-204].
Reviewer Comment 3: In case where this manuscript is still within the review article, I suggest to delete all new result of re-analysis done by the authors and to summarize points to deliver to the readership. (I guess their message is within 'canonical pathways' in ASD.)
The categorization of this manuscript as a review rather than an original research manuscript was carefully considered by the authors before submission, with the specific aim and scope of the manuscript in mind. It was not our objective to collate raw datasets, or to bioinformatically combine and reanalyse them to present new findings. Instead, we collated and compared previously existing datasets with respect to their functional enrichment signatures.
The focus of our manuscript was to review the canonical pathways in ASD molecular data, and we used a quantitative approach to do this. This approach allowed us to discuss and compare different datasets with respect to standardised functional enrichment signatures instead of integrating the findings that were reported across different studies that each used different approaches to functionally annotate them.
We chose Hallmark “canonical pathways” to functionally annotate each dataset specifically because these datasets were developed in order to reduce the heterogeneity and redundancy present in large datasets such as MSigDB. Each Hallmark gene set represents a well-defined biological pathway or process in one refined and curated gene set, derived from multiple founder gene sets. Hallmark gene sets are particularly useful for GSEA, as they summarize the information of many founder sets while reducing variation and redundancy among them.
While we could summarise our findings without including a detailed section on methodology and results, we consider that part of the novelty of this review is that it identifies shared canonical pathways in the current ASD literature based on a quantifiable approach. We consider this a valuable approach to review different ASD datasets due to the heterogeneity of this specific disorder. In the context of ASD, different datasets from different cohorts might implicate disparate genes and proteins due to the complex genetic architecture and aetiological heterogeneity of the disorder. These disparate gene sets might share similar functional enrichment signatures that could point to convergent underlying mechanisms in ASD. Therefore, our review does not aim to report original findings, nor to demonstrate a clear link between molecular mechanisms and disease aetiology; we have made this more clear in our revised submission [see lines 357 – 360]. Our review focuses on summarising the current evidence for commonly implicated mechanisms in ASD in the context of what is currently known about ASD aetiology.
Reviewer 2 Report
The review by Mahony and Ryan utilizes multiple -omics analyses (transcriptomics, methylomics, proteomics) of clinically defined sets of individuals with ASD and controls to identify canonical pathways underlying ASD. The review is well written and interesting and is a critical evaluation as these sets of data often don’t overlap and identifying overlapping pathways is interesting. There are two minor areas that could be addressed to strengthen the review. However, this is an important, well-written, and interesting appraisal of different ASD omics studies.
- The first table of the studies is critical, but it the formatting of the table makes it difficult to see which studies go with which collection type (blood, brain, etc) and omic class (transcriptomic). Reformatting the table would be helpful and breaking out the 3 omics and spreading the table out for readability would help alleviate this issue.
- Additional discussion on caveats of merging different omics approaches should be given. Synaptogenesis and neurogenesis are mentioned, but the number of samples from the proteomics studies that are obtained during these periods are limited (neurogenesis and synaptogenesis occurring at young ages and the brain tissue age range being 2-60 years of age – it is unclear how many people fit that range. Also, the protein expression may be different across different brain regions and the regions collected for these studies are limited. These are challenges with any human, proteomic, post-mortem study, but these limitations may inform why there is less overlap b/w the proteomic and other studies which makes sense. Mentioning this and that there may be a stronger difference between individuals depending on the age could be mentioned.
Mistype:
Line 342 says “with both in vitro - [118] and in vitro- studies [119] showing. . .” Should one of these be in vivo?
Author Response
We thank the reviewer for acknowledging the strengths of our paper. The reviewer’s comments are insightful and constructive. We have incorporated all the reviewer’s comments into our revised manuscript, as described in detail below:
Reviewer Comment 1: The first table of the studies is critical, but it the formatting of the table makes it difficult to see which studies go with which collection type (blood, brain, etc) and omic class (transcriptomic). Reformatting the table would be helpful and breaking out the 3 omics and spreading the table out for readability would help alleviate this issue.
We have reformatted this table to clearly differentiate between different molecular approaches and different tissue types [see Table 1].
Reviewer Comment 2: Additional discussion on caveats of merging different omics approaches should be given.  Synaptogenesis and neurogenesis are mentioned, but the number of samples from the proteomics studies that are obtained during these periods are limited (neurogenesis and synaptogenesis occurring at young ages and the brain tissue age range being 2-60 years of age – it is unclear how many people fit that range. Also, the protein expression may be different across different brain regions and the regions collected for these studies are limited. These are challenges with any human, proteomic, post-mortem study, but these limitations may inform why there is less overlap b/w the proteomic and other studies which makes sense. Mentioning this and that there may be a stronger difference between individuals depending on the age could be mentioned.  
We have added additional discussion on the topics recommended by the reviewer. We have added an additional paragraph to outline the limitations of the proteomics datasets, particularly with respect to the implications for neurodevelopment [see lines 165 – 167; 348 – 353]. Where possible, we have amended Table 1 to include information about the average age of individuals across different cohorts, and we have highlighted relevant aspects of this information in our results [see lines 142 - 183]. We have also highlighted the fact that protein expression varies widely between different brain regions [see lines 294-299]. Additionally, we note that the studies that we included in our review largely report data from either the prefrontal cortex or the temporal cortex. These regions are relevant in the context of ASD, given that previous studies have found that ASD dysregulated features were predominantly localized to the cerebral cortex (Ramaswami et al, 2020).
We have also added an explanation of how the canonical pathways implicated in proteomic data share common upstream regulators with the canonical pathways implicated using other molecular methods. Hence, our discussion on how these pathways are convergent across different datasets [see lines 323 – 329].
We have clarified the limitations of our results in the context of neurodevelopment given that the datasets used are generally not representative of early embryogenesis [see lines 348 – 360]. We have clarified that, given the absence of databanks or tissue samples that are derived from ASD individuals during neurodevelopment, the functional implications for ASD omics datasets only provide limited insight into the mechanisms that might be dysregulated during neurodevelopment. We have highlighted how this limitation accentuates the utility of the integration of functional enrichment signatures from a wide range of tissue types and individuals of different ages. Our discussion highlights canonical pathways that are dysregulated across many different tissue types at many different ages. We have clarified that we do not aim to demonstrate a direct link between these molecular signatures and mechanisms in ASD aetiology, but rather that these common pathways may point to common underlying mechanisms in ASD, which has potential relevance in the context of neurodevelopment.
Reviewer Comment 3: Mistype: Line 342 says “with both in vitro - [118] and in vitro- studies [119] showing. . .” Should one of these be in vivo?
This mistype has been corrected in the revised submission [line 431].
Round 2
Reviewer 1 Report
I appreciate the authors' work and am satisfied with the revised manuscript.